# Importance of Timely Treatment Initiation in Infantile-Onset Pompe Disease, a Single-Centre Experience

**DOI:** 10.3390/children8111026

**Published:** 2021-11-09

**Authors:** Javier de las Heras, Ainara Cano, Ana Vinuesa, Marta Montes, María Unceta Suarez, Arantza Arza, Saioa Jiménez, Elena Vera, Marta del Hoyo, Miriam Gendive, Lizar Aguirre, Gisela Muñoz, Javier Fernández, Cynthia Ruiz-Espinoza, María Ángeles Fernández, José Miguel Galdeano, Irene Rodríguez, Lourdes Román, Amaya Rodríguez-Serna, Begoña Loureiro, Itziar Astigarraga

**Affiliations:** 1Division of Pediatric Metabolism, CIBER-ER, Cruces University Hospital, 48903 Barakaldo, Spain; ana.vinuesajaca@osakidetza.eus (A.V.); martamontes8@outlook.es (M.M.); 2Biocruces Bizkaia Health Research Institute, 48903 Bizkaia, Spain; ainara.canosanjose@osakidetza.eus (A.C.); maria.uncetasuarez@osakidetza.eus (M.U.S.); arantzazu.arzaruesga@osakidetza.eus (A.A.); giselacristina.munozgarcia@osakidetza.eus (G.M.); javier.fernandezaracama@osakidetza.eus (J.F.); cynthializ.ruizespinoza@osakidetza.eus (C.R.-E.); mariaangeles.fernandezcuesta@osakidetza.eus (M.Á.F.); josemiguel.galdeanomiranda@osakidetza.eus (J.M.G.); lourdes.romanechevarria@osakidetza.eus (L.R.); amaya.rodriguezserna@osakidetza.eus (A.R.-S.); begona.loureirogonzalez@osakidetza.eus (B.L.); mariaiciar.astigarragaaguirre@osakidetza.eus (I.A.); 3Department of Pediatrics, Cruces University Hospital, 48903 Barakaldo, Spain; 4Department of Pediatrics, University of the Basque Country (UPV/EHU), 48940 Leioa, Spain; elena.veradepedro@osakidetza.eus; 5Metabolism Section, Biochemistry Laboratory, Cruces University Hospital, 48903 Bizkaia, Spain; 6Department of Pediatrics, Araba University Hospital, 01009 Vitoria, Spain; saioa.jimenezechevarria@osakidetza.eus (S.J.); marta.delhoyomoracho@osakidetza.eus (M.d.H.); miriam.gendivemartin@osakidetza.eus (M.G.); lizar.aguirrepascasio@osakidetza.eus (L.A.); 7Department of Pediatrics, Basurto University Hospital, 48013 Bilbao, Spain; 8Department of Cardiology, Cruces University Hospital, 48903 Barakaldo, Spain; rodrigueztoirene@gmail.com

**Keywords:** Pompe disease, infantile Pompe disease, immune tolerance induction, immunomodulation, early diagnosis, enzyme replacement therapy

## Abstract

Classic infantile Pompe disease (IPD) is a rare lysosomal storage disorder characterized by severe hypertrophic cardiomyopathy and profound muscle weakness. Without treatment, death occurs within the first 2 years of life. Although enzyme replacement therapy (ERT) with alglucosidase alfa has improved survival, treatment outcome is not good in many cases and is largely dependent on age at initiation. The objective of the study was (a) to analyse the different stages in the diagnosis and specific treatment initiation procedure in IPD patients, and (b) to compare clinical and biochemical outcomes depending on age at ERT initiation (<1 month of age vs. <3 months of age). Here, we show satisfactory clinical and biochemical outcomes in two IPD patients after early treatment initiation before 3 months of life with immunomodulatory therapy in the ERT-naïve setting, with a high ERT dose from the beginning. Despite the overall good evolution, the patient who initiated treatment <1 month of life presented even better outcomes than the patient who started treatment <3 months of life, with an earlier normalization of hypertrophic cardiomyopathy, along with CK normalization, highlighting the importance of early treatment initiation in this progressive disease before irreversible muscle damage has occurred.

## 1. Introduction

Pompe disease (OMIM #232300), also known as glycogenosis type II, is a rare autosomal recessive lysosomal storage disorder caused by a deficiency of the lysosomal enzyme acid alpha-glucosidase (GAA). It is characterized by the abnormal accumulation of lysosomal glycogen in skeletal, cardiac, bulbar and smooth muscle, leading to myopathy, respiratory weakness, physical disability and premature death [1]. Classic infantile Pompe disease (IPD) is the most severe form and presents with hypertrophic cardiomyopathy, which can be detected even prenatally [2], as well as profound muscle weakness in the first months of life, causing severely delayed motor development and compromised respiratory function. Without treatment, death occurs within the first two years of life, usually as a result of cardiorespiratory failure [3]. Enzyme replacement therapy (ERT) with recombinant human acid alpha-glucosidase (rhGAA), alglucosidase alfa, has been commercially available since 2006 and has led to improvements in survival rates and clinical outcomes [4,5]. However, clinical outcomes are not optimal, with many IPD patients treated with ERT presenting severe hypotonia requiring respiratory and nutritional support such as invasive ventilation and G-tube, and inability to walk [4,6]. Treatment outcome is largely dependent on age at initiation, making it crucial to commence before irreversible muscle damage has occurred. Moreover, it has been shown that IPD patients who start ERT <3 months of age have better survival rates and better clinical outcomes than patients who start ERT ≥ 3 months of age [7]. Therefore, prompt diagnosis and urgent treatment initiation is crucial for IPD patients. The objective of the present study is a) to analyse the different stages in the diagnosis and specific treatment initiation procedure in IPD patients followed at the Metabolic Unit at Cruces University Hospital, Spain, and b) to compare clinical and biochemical outcomes depending on age at ERT initiation (<1 month of age vs. <3 months of age).

## 2. Patients and Methods

The study protocol was performed according to the ethical guidelines of the revised 1975 Declaration of Helsinki [8] and approved by the Research Ethics Committee of the Basque Country (CEIm-E), ethical approval code: PI2021168. Written informed consent was obtained from the parents of the study participants.

The inclusion criteria for the study were: (a) established diagnosis of IPD within the first three months of life confirmed by both GAA enzyme activity testing and genetic mutational analysis; (b) regular attendance to their scheduled clinical follow-up visits at the Metabolic Unit at Cruces University Hospital, Spain; and (c) willingness to participate in the study.

Two patients fulfilled the inclusion criteria: patient 1 started specific treatment at 2 months and 20 days of age (<3 months of age) and patient 2 at 18 days of age (<1 month of age). Both patients received exactly the same treatment regimen, with prophylactic immune tolerance induction (ITI) and ERT with rhGAA. The ITI regimen included four doses of weekly rituximab (375 mg/m^2^, intravenously), three cycles of methotrexate (0.4 mg/kg; three doses per cycle with the first three ERT infusions, subcutaneously or orally), and monthly intravenous immunoglobulin (IG) (400 mg/kg) for a period of 6 months, which was initiated along with ERT, as recently described by Desai et al. [9]. In this protocol, rituximab is administered prior to the first IG infusion to avoid a potential rituximab half-life reduction due to a previous IG infusion. Both patients have received ERT with alglucosidase alfa at a dose of 40 mg/kg every other week since treatment initiation (Figure 1).

In this retrospective review of clinical cases, data on the timing of the different steps in the diagnostic & treatment initiation procedure was collected from the clinical records. In order to compare the evolution depending on the age at specific treatment initiation (<1 month of age vs. <3 months of age), retrospective longitudinal data on clinical assessments of efficacy and biochemical parameters were also collected from medical records.

For clinical assessments of efficacy, left ventricular mass index (LVMI) (g/m^2^) were performed using two-dimensional, M-mode and doppler echocardiography, and motor development was evaluated using the Alberta Infant Motor Scale (AIMS) [10].

## 3. Results

### 3.1. Presentation of Clinical Cases

#### 3.1.1. Patient 1

Patient 1 was born to non-consanguineous parents at 37 weeks of gestational age with normal birth weight (3.61 kg). At 2 months of age, a heart murmur and subtle hepatomegaly (that have not been noted again since) were observed. On further investigation, CK and hepatic transaminases were elevated (CK: 515 U/L; ALT: 85 U/L; AST: 145 U/L), and an echocardiogram showed severe hypertrophic cardiomyopathy. The child was asymptomatic, with normal neurological examination, and did not present macroglossia or hypotonia. Dried blood spot (DBS) & peripheral blood lymphocyte culture GAA analysis showed significantly reduced GAA enzyme activity [DBS GAA = undetectable; blood lymphocyte GAA = 0.1 nmol/h/mg (normal range: 1.8 to 11.3)]. *GAA* next generation sequencing (NGS) showed two variants in compound heterozygosis, the previously described c.573C>A (p.Tyr191*) (variant classification: substitution; predicted severity: very severe; predicted CRIM status: negative), and the novel variant c.1075+2T>C.

#### 3.1.2. Patient 2

Patient 2 was born to non-consanguineous parents of Nepali origin at 41 weeks of gestational age with normal birth weight (3.6 kg). On his first day of life, he was admitted to the neonatal ward due to transitory respiratory distress and unconjugated hyperbilirubinemia. On examination, a heart murmur was heard and echocardiography revealed severe hypertrophic cardiomyopathy. He presented elevated CK and slightly elevated ALT levels (CK: 1146 U/L; ALT: 59 U/L), with AST within the normal range (AST: 115 U/L). The child was asymptomatic, with normal neurological examination, and did not present macroglossia, hypotonia or hepatomegaly. IPD was confirmed with reduced GAA activity (DBS GAA = 0.7 μmol/L/h (normal range: 7 to 37); blood lymphocyte GAA = undetectable). Molecular analysis by NGS showed that the patient presented three GAA sequence variants, two inherited from his father, c.505C>A (p.Leu169Met) (variant classification: substitution; predicted severity: potentially less severe; predicted CRIM status: positive) and c.1636+5G>A (p.?) (variant classification: substitution; predicted severity: very severe; predicted CRIM status: unknown); and one inherited from his mother, c.1579_1580del (p.Arg527Glyfs*3) (variant classification: deletion; predicted severity: very severe; predicted CRIM status: unknown).

### 3.2. Different Stages of the Diagnostic and Specific Treatment Initiation Procedure 

The different chronological steps of the diagnostic and treatment commencement procedure are depicted in Table 1. In both patients, time from first severe hypertrophic cardiomyopathy observation to specific treatment initiation was very similar (18 and 17 days).

### 3.3. Clinical & Biochemical Outcomes by Age at Treatment Initiation (<3 Months of Age vs. <1 Month of Age)

#### 3.3.1. Cardiac Status

Both patients presented severe hypertrophic cardiomyopathy at baseline. The baseline LVMI was 287 g/m^2^ in Patient 1 and 216 g/m^2^ in Patient 2, the upper control limit for infants being 64 g/m^2^ [11]. Figure 2 shows longitudinal LVMI evolution after ERT treatment commencement. Hypertrophic cardiomyopathy resolved earlier in the patient who started ERT before 1 month of age (12 weeks vs. 20 weeks after treatment initiation). 

#### 3.3.2. Motor Development and Ambulation

Both patients presented longitudinal determinations of AIMS within the normal range. Patient 1, who started ERT < 3 months of age, scored between the 5th and 10th percentile in all determinations between 4 and 16 months of age. However, Patient 2, who started ERT < 1 month of age, scored slightly above the 50th percentile at baseline, and measured between the 75th and 90th percentile in the most recent assessment at 9 months of age. Developmentally, Patient 1 was sitting by 8 months of age and walked independently at 15 months of age. Patient 2 walked independently at 10.5 months of age.

#### 3.3.3. Growth, Respiratory and Gastrointestinal Status

Both patients have presented normal growth, following their typical curves for height and weight, within the normal range for age and sex. Both patients have been asymptomatic, not requiring any respiratory support, and eating an age-appropriate diet by mouth with no assistance, and none of them has suffered any significant respiratory infection.

#### 3.3.4. Biochemical Parameters

##### CK and Hepatic Transaminase Levels

Patient 1, who started ERT < 3 months of age, presented elevated CK and AST & ALT levels at baseline, which have remained persistently high. However, Patient 2, who started ERT < 1 month of age, presented elevated CK and ALT levels at baseline, with AST within the normal range. CK levels normalized 8 weeks after treatment commencement, and he still presents slightly raised ALT levels (Figure 3).

##### Urinary Glucose Tetra-Saccharide (Glc4)

The evolution of urinary Glc4 levels is shown in Figure 4. Patient 1, who started specific treatment < 3 months of age, normalized urine Glc4 levels 4 weeks after treatment initiation, and presented elevated urine Glc4 levels again at the most recent determination (at 75 weeks after ERT initiation). Patient 2, who started specific treatment < 1 month of age, has presented all Glc4 levels within the normal range (Figure 4).

#### 3.3.5. Anti-rh GAA IgG Antibodies

Anti-alglucosidase alfa IgG antibodies have not been detected in either of the two patients, with the most recent determination in Patient 1 being carried out at 17 months after ERT initiation, and in Patient 2 at 7 months after treatment initiation.

## 4. Discussion

Clinical features in classic Pompe disease (hypotonia, muscle weakness, respiratory insufficiency, feeding difficulties and cardiac problems) are non-specific, and diagnostic delays are common, with a 3-month delay on average after the onset of symptoms [13]. This is particularly important because treatment outcome is largely dependent on age at initiation, and it is crucial to commence ERT before irreversible muscle damage has occurred, which means that ERT should be initiated as soon as possible [14], as suggested by the evolution of the two patients of this study. 

In areas where new-born screening (NBS) for Pompe disease is not implemented, prompt diagnosis relies on high clinical awareness. Furthermore, as specific treatment should ideally be initiated before clinical symptoms appear, serendipity may play an important role, as is the case in our patients. Our two patients were asymptomatic when an echocardiogram was performed after a heart murmur was heard and they were in a hospital setting because of a banal disease.

In our patients, from two different hospitals without a specific Metabolic Unit, once severe hypertrophic cardiomyopathy was observed, clinical suspicion was aroused and our unit, a referral Metabolic Unit at a tertiary hospital, was immediately contacted. As some physicians who are not familiar with IPD may recognize the disease as a combination of cardiomyopathy and hypotonia, it is important to emphasize that a diagnosis of IPD cannot be excluded because of the absence of the latter. In fact, neither of our patients presented hypotonia, and it is precisely before hypotonia appears that ERT should be started.

A diagnostic odyssey, or the journey of the patients from the first contact with a health care provider to the expert centre, is common in rare diseases and, as such, in Pompe disease [15]. In our two cases this contact was straightforward, due in part to the fluid relationship between our Metabolic Unit and the different health care providers in our referral area (other hospitals and also primary care centres), as we usually run educational activities on inborn errors of the metabolism.

A median time of 0.5 months has been reported from genetic and cross-reactive immunological material (CRIM) status diagnosis to commencement of ERT treatment, primarily due to administrative or regulatory issues (i.e., procurement of resources) in centres with expertise in Pompe disease [6]. In our case, as early treatment initiation is vital, once IPD suspicion was very high (severe hypertrophic cardiomyopathy, high CK & transaminase levels and very low DBS GAA activity), administrative procedures were initiated, and so specific treatment could be initiated as soon as genetic confirmation was received. At this stage, an urgent meeting of the Pharmacy Committee that approves these special treatments in rare diseases was set up, and rapid approval was obtained. Surgery for intravenous catheter placement was also scheduled. Once the diagnosis was confirmed, a central venous catheter was placed and specific treatment was initiated.

As a result, diagnosis confirmation and specific treatment initiation were performed quite fast in our two patients, Patient 1 starting at 2 months and 20 days of age, and Patient 2 at 18 days of age. In fact, our second patient started specific treatment even earlier than a recently reported child diagnosed by NBS who started at 21 days of life [16].

Among patients with Pompe disease, CRIM status is an important predictor of response to ERT. In the absence of a prophylactic ITI treatment, CRIM-negative patients, who have two deleterious mutations and no GAA protein expression, usually develop high sustained anti-rhGAA antibody titers (HSAT), which lead to an abrupt clinical decline that is often fatal [17]. Although CRIM-positive patients, with missense mutations and some residual GAA protein, usually do not develop HSAT, there are some exceptions which also result in clinical decline and death [18]. Prophylactic ITI treatment is generally recommended for CRIM-negative IPD patients, as it has shown great success in inducing immune tolerance in ERT-naïve IPD patients [6,19]. As ITI treatment has been shown to be safe [6], in order to start ERT as soon as possible and to not take the risk, albeit low, of developing HSAT in a CRIM-positive patient, we decided to treat our two patients immediately after the diagnosis was made, using immunomodulation therapy in the ERT-naïve setting irrespective of CRIM status, as recently suggested by Owens et al. [18]. Our patients have tolerated ITI without any adverse events. It has been reported that most IPD patients develop anti-rhGAA antibodies at a median age of 1.8 months in the absence of ITI treatment [20]. Another study in IPD patients who received prophylactic ITI treatment showed that 4/7 patients remained antibody free, and 3/7 patients developed antibodies at 23, 31 and 39 weeks after treatment initiation, two of them requiring another course of the ITI regimen [6]. Our patients have not developed anti-drug antibodies so far.

As patients with infantile-onset Pompe disease suffer from marked hypertrophic cardiomyopathy and an increased risk of arrhythmia, and consequently may present a high anaesthetic risk, we wondered whether a totally implantable central venous port system should be placed before ITI & ERT initiation or later, when hypertrophic cardiomyopathy would have improved after some infusions of ERT. As it is reported that deaths resulting from arrhythmia during anaesthetic procedures in children with IPD correlated with LVMI > 350 g/m^2^ [21], and the ITI & ERT regimen comprised multiple intravenous infusions and blood extractions particularly during the first month of treatment (Figure 1), the decision was made to place the implantable venous catheter before specific treatment was initiated, following the anaesthetic recommendations for patients with IPD, detailed in the paper by Wang et al. [21].

Recently it has been reported that higher dosing of alglucosidase alfa is safe and improves outcomes in children with Pompe disease [22,23]. As suggested by Chien et al. [22], we decided to treat our two patients with a high dose of alglucosidase alfa (40 mg/kg every other week) as the starting dose, instead of waiting for a potential clinical decline to raise the dosage, as muscle damage is irreversible in this rapidly progressive disease. Neither of our two patients has experienced any infusion-associated reactions to the ERT to date.

The diagnosis of Pompe disease is usually established by demonstrating a deficiency in GAA enzyme activity and finding disease-causing mutations using molecular analysis of the *GAA* gene. The finding of one known mutation and a sequence variant, a DNA change with unproven pathogenicity, or two sequence variants of possible pathogenicity requires confirmation demonstrating low GAA activity, or histological evidence from a muscle biopsy [1]. The Pompe disease *GAA* variant database (http://www.pompevariantdatabase.nl) is an open-source Pompe disease-specific database, and lists disease-associated *GAA* variants, along with in silico predictions, clinical phenotypes and predicted CRIM status for these disease-associated variants [24]. As we decided to use ITI treatment irrespective of CRIM status, western blot analysis was not performed, and CRIM status based on the genotype was undetermined in both patients [24].

Both patients have shown a satisfactory evolution as they have remained asymptomatic so far, presenting normal development and scoring within the normal range in AIMS, in addition to not requiring any respiratory or nutritional assistance, while a certain percentage of IPD patients treated with ERT still presents severe hypotonia requiring respiratory and nutritional support, as well as walking incapability [4,6]. The better outcome observed in our patients may be due to the early treatment initiation, the immunomodulation therapy in the ERT-naïve setting, and the high ERT dose from the beginning. Despite the overall good evolution of the patients, in this study we show that ERT should be initiated as soon as possible, as the patient who initiated treatment before the first month of life presented an even better evolution than the patient who started treatment before the third month of life, with an earlier normalization of hypertrophic cardiomyopathy, along with CK normalization, which is not frequent among IPD survivors [20].

The main limitation of the present study is sample size, which is due to the rare nature of the disease, with a very limited number of patients with a diagnosis of IPD available. However, we believe that two IPD patients from a single centre with exactly the same treatment regimen and the same clinical & biochemical follow-up is a fairly good number. We decided to perform a single-centre study to avoid the heterogenicity of treatment and follow-up regimens in different centres, which would have made it difficult to draw conclusions. We also acknowledge that, besides age at treatment initiation, differences in treatment outcomes may be influenced by other factors such as the genotype, CRIM status, or residual enzymatic activity. Although in this study we present short-term follow-up, differences in the evolution of the two patients can be appreciated. Nevertheless, long-term narrow follow-up is warranted.

## 5. Conclusions

In this study, we show satisfactory clinical and biochemical outcomes in two IPD patients after early treatment initiation before three months of life with immunomodulation therapy in the ERT-naïve setting, and high ERT dose from the beginning. Furthermore, the patient who initiated treatment before the first month of life presented even better evolution than the patient who started treatment before the third month of life, highlighting the importance of early treatment initiation in this progressive disease, before irreversible muscle damage has occurred.

## Figures and Tables

**Figure 1 children-08-01026-f001:**
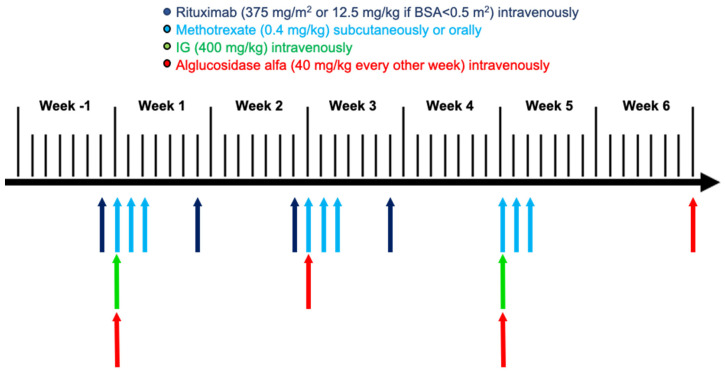
Scheme of the first 6 weeks of the treatment regimen used for the study’s participants, which includes immune tolerance induction along with enzyme replacement therapy. Body surface area (BSA), immunoglobulin (IG).

**Figure 2 children-08-01026-f002:**
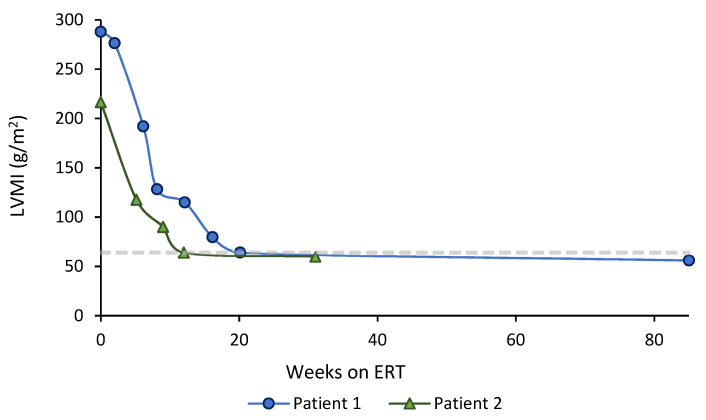
Left ventricular mass index (LVMI) evolution after ERT initiation in Patient 1 (started ERT < 3 months of age) (blue line), and Patient 2 (started ERT < 1 month of age) (green line). The horizontal dashed line indicates the upper limit of normal for LVMI (g/m^2^) in healthy infants (64 g/m^2^) [11].

**Figure 3 children-08-01026-f003:**
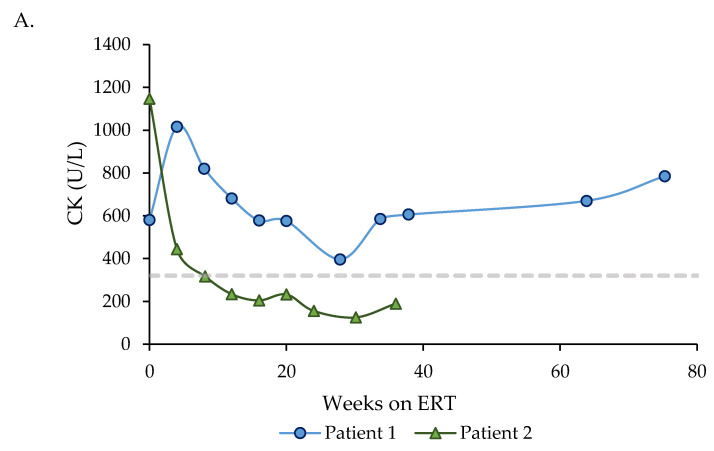
Evolution of serum CK, AST & ALT levels after ERT initiation in Patient 1 (started ERT < 3 months of age) (blue lines), and Patient 2 (started ERT < 1 month of age) (green lines). (**A**) Evolution of serum CK levels. The horizontal dashed line indicates the upper limit of the normal range (320 U/L, age 0 to 9 years). (**B**) Evolution of serum AST levels. The horizontal dashed line indicates the upper limit of the normal range (120 U/L, age < 12 months; 41 U/L, age > 12 months). (**C**) Evolution of serum ALT levels. The horizontal dashed line indicates the upper limit of the normal range (45 U/L, age < 12 months; 40 U/L, age 1 to 11 years).

**Figure 4 children-08-01026-f004:**
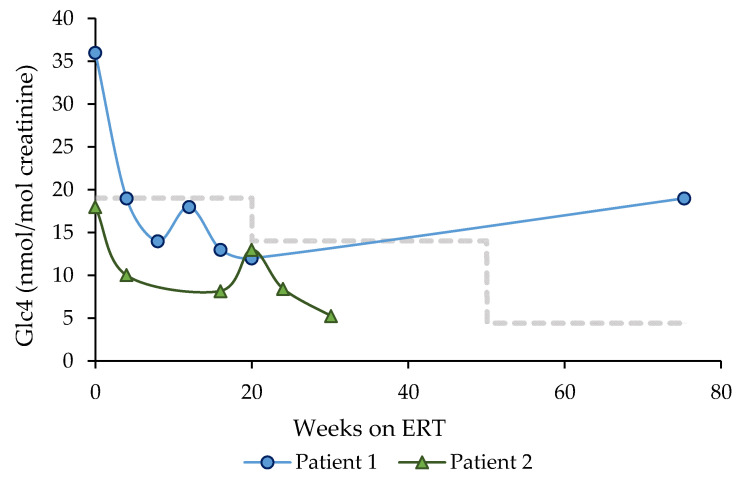
Evolution of urinary glucose tetra-saccharide (Glc4) levels after ERT initiation in Patient 1 (started ERT < 3 months of age) (blue line), and Patient 2 (started ERT < 1 month of age) (green line). The horizontal dashed line indicates the upper limit of the control range (19 mmol/mol creatinine, age 0 to 6 months; 14.0 mmol/mol creatinine, age 6 to 12 months; 4.4 mmol/mol creatinine, age > 12 months of age [12]). Enzyme replacement therapy (ERT).

**Table 1 children-08-01026-t001:** Timing at the different stages of the diagnostic and specific treatment initiation procedure. Dried blood spot (DBS); acid alpha-glucosidase (GAA); immune tolerance induction (ITI); enzyme replacement therapy (ERT).

	Patient 1 (Age)	Patient 2 (Age)
1-Severe hypertrophic cardiomyopathy first observed	2 months and 2 days	1 day
2-Reference Metabolic Unit contacted	2 months and 3 days	4 days
3-DBS GAA enzyme analysis result	2 months and 10 days	7 days
4-Treatment request to Pharmacy Committee	2 months and 13 days	7 days
5-Peripheral blood lymphocyte culture GAA analysis result	2 months and 14 days	12 days
6-Genetic testing result	2 months and 16 days	14 days
7-Totally implantable central venous port system placement	2 months and 17 days	17 days
8-ITI & ERT regimen initiation	2 months and 20 days	18 days

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
