# Peer review of "Importance of Timely Treatment Initiation in Infantile-Onset Pompe Disease, a Single-Centre Experience"

_children, 2021, doi:10.3390/children8111026_

Round 1

Reviewer 1 Report

The proposes of this manuscript are to demonstrate the importance of early treatment of IPD and the process of diagnosis and treatment initiation at their center. The authors presented the timeline from identifying IPD patients to the treatment initiation as well as compared the clinical and biochemical outcomes when the treatment with ITI and ERT were started before vs after 3 months of age.

The authors stated that both IPD patients received the same treatment including same regimen of ITI and ERT dosage, but the only difference was that one patient received the treatment a few months earlier than another patient. The turnaround time of genetic test result is very impressive, only 6-7 days, which is faster than in general especially in the centers which do not have molecular genetic lab in house. Although CRIM status was not determined/mentioned prior to ITI initiation, patients received the prophylactic ITI regimen which normally is used for CRIM negative patients without developing adverse reactions. The authors mentioned using the central line for management. For infants with HCM who require long term ERT, central line should be placed and treatment should be conducted in ICU setting. Port A Cath such as a miniport is definitely the option when the conditions are stable.

Comments

  1. Introduction line 63-64, consider removing redundant information “Although ERT with alglucosidase alfa has greatly improved survival in IPD”
  2. Results:
    1. variant classification and the predicted severity as well as predicted CRIM status should be mentioned with the results of molecular genetic testing. Patient 1 is a compound heterozygous of pathogenic variant c.573C>A (p.Tyr191*) and C.1075+2T>C. the classification of the second variant was not mentioned. Is this variant classified as VUS? c.573C>A is predicted to cause severe phenotype-classic IPD and CRIM negative. If the variant classification and predicted phenotype mentioned in results, the first sentence in page 9 can be removed.
    2. How long were both patients followed? What were their ages? From the figure, it looks like patient 1 had been followed longer than patient 2.
    3. Comparison clinical and biochemical outcomes of patient 1 vs patient 2, in summary it took longer for patient 1’s HCM and biomarkers to be normalized (or almost normalized). Are those parameters have any clinical impact? HCM was finally normal although it took a bit longer. Does near normal CK have any clinical impact? Patient 1 walked independently at 15 months of age while patient 2 walked independently at 10.5 months of age but both had AIMS in the normal ranges, so no clinically significance?
    4. Growth and number of respiratory tract infections can also be used to determine the outcomes.
  3. Discussion:
    1. CRIM status is one of the important factors to determine the outcomes. Based on the genotypes, patient 1 was likely having CRIM negative and CRIM status of patient 2 was undetermined. Although I agreed that the resolution of patient’s HCM and biomarkers were better than patient 1’s, the fact that their genotypes were different should be mentioned as one of the limitations of this study.
    2. Another limitation of this study, long term follow up is important since some patients who received prophylactic ITI and high dose ERT in the newborn or infancy periods may develop mild muscle weakness especially facial muscle in childhood period.
    3. Please clarify “A median time of 0.5 months has been reported from genetic and cross-reactive immunological material (CRIM) status” in line 225-227, what type of the study did the author refer to? Western blot CRIM study?
    4. Line 254, the authors compared the anti-rhGAA antibodies after ITI of their 2 patients by using data from ref #20. Did the study group from ref#20 receive ITI? In theory, patients who received ITI should not develop high anti-rhGAA antibodies.
    5. To compare the outcomes of these 2 patients with the literature, besides age of patients when the treatment starts, the other factors should be comparable e.g. CRIM status, ERT dosage (40MK weekly vs EOW vs 20MK), prophylactic ITI vs therapeutic ITI and whether the patients developed anti-rhGAA antibodies after ITI.

Author Response

The proposes of this manuscript are to demonstrate the importance of early treatment of IPD and the process of diagnosis and treatment initiation at their center. The authors presented the timeline from identifying IPD patients to the treatment initiation as well as compared the clinical and biochemical outcomes when the treatment with ITI and ERT were started before vs after 3 months of age.

The authors stated that both IPD patients received the same treatment including same regimen of ITI and ERT dosage, but the only difference was that one patient received the treatment a few months earlier than another patient. The turnaround time of genetic test result is very impressive, only 6-7 days, which is faster than in general especially in the centers which do not have molecular genetic lab in house. Although CRIM status was not determined/mentioned prior to ITI initiation, patients received the prophylactic ITI regimen which normally is used for CRIM negative patients without developing adverse reactions.

Thank you very much for your review and insightful comments. We have answered all your questions/comments on a point-by-point reply and believe that our manuscript is much improved with the incorporation of the suggested comments. Our point-by-point response to each comment is detailed below.

Point 1.

The authors mentioned using the central line for management. For infants with HCM who require long term ERT, central line should be placed and treatment should be conducted in ICU setting. Port A Cath such as a miniport is definitely the option when the conditions are stable.

Response 1.

Thank you for pointing out this issue. We have had a problem with the translation of the term. In fact, a totally implantable central venous port system was implanted in both patients. We have changed the term along the text.

Point 2.

Introduction line 63-64, consider removing redundant information “Although ERT with alglucosidase alfa has greatly improved survival in IPD”

Response 2.

The text has been shortened according to the reviewer´s suggestion.

RESULTS

Point 3.

1.variant classification and the predicted severity as well as predicted CRIM status should be mentioned with the results of molecular genetic testing. Patient 1 is a compound heterozygous of pathogenic variant c.573C>A (p.Tyr191*) and C.1075+2T>C. the classification of the second variant was not mentioned. Is this variant classified as VUS? c.573C>A is predicted to cause severe phenotype-classic IPD and CRIM negative. If the variant classification and predicted phenotype mentioned in results, the first sentence in page 9 can be removed.

Response 3.

As suggested, variant classification, predicted severity and predicted CRIM status of every variant has been added to the results of molecular testing:

Patient 1

GAA next generation sequencing (NGS) showed two variants in compound heterozygosis, the previously described c.573C>A (p.Tyr191*) (variant clasification: substitution; predicted severity: very severe; predicted CRIM status: negative), and the novel variant c.1075+2T>C.

Patient 2

Molecular analysis by NGS showed that the patient presented three GAA sequence variants, two inherited from his father, c.505C>A (p.Leu169Met) (variant clasification: substitution; predicted severity: potentially less severe; predicted CRIM status: positive) and c.1636+5G>A (p.?) (variant clasification: substitution; predicted severity: very severe; predicted CRIM status: unknown); and one inherited from his mother, c.1579_1580del (p.Arg527Glyfs*3) (variant clasification: deletion; predicted severity: very severe; predicted CRIM status: unknown).

Although CRIM status can be determined in 90% patients based on their genotype, unfortunately CRIM status is undetermined in our two patients.

Patient 1 presents one novel variant with no information on predicted CRIM status.

And Patient 2 presents 3 variants in two alleles (2 variants in one allele and one variant in the other allele). There is one variant in each allele with no information on predicted CRIM status, therefore CRIM status cannot be determined.

As suggested, the first sentence in page 9 has been removed, and the following sentence has been added: “As we decided to use ITI treatment irrespective of CRIM status, western blot analysis was not performed, and CRIM status based on the genotype was undetermined in both patients.”

Point 4.

How long were both patients followed? What were their ages? From the figure, it looks like patient 1 had been followed longer than patient 2.

Response 4.

Patient 1 was born on 2019 and we present follow-up until 22 months of age (85 weeks after treatment initiation).

Patient 2 was born on 2020 and we present follow-up until 11 months of age (44 weeks after treatment initiation).

Point 5.

Comparison clinical and biochemical outcomes of patient 1 vs patient 2, in summary it took longer for patient 1’s HCM and biomarkers to be normalized (or almost normalized). Are those parameters have any clinical impact? HCM was finally normal although it took a bit longer. Does near normal CK have any clinical impact? Patient 1 walked independently at 15 months of age while patient 2 walked independently at 10.5 months of age but both had AIMS in the normal ranges, so no clinically significance?

Response 5.

There is an important difference in CK values between the two patients. Despite her high CK values, Patient 1 is presenting a normal motor development so far, but probably will present some mild symptoms in the future, and close follow-up is warranted.

Although Patient 2 walked independently at 10.5 months of age and Patient 1 at 15 months of age, as they both scored within the normal range in AIMS, we agree with the reviewer´s comment that there is not a clinical significance and state accordingly in the text: “Both patients have shown a satisfactory evolution as they have remained asymptomatic so far, presenting normal development and scoring within the normal range in AIMS”.

Also, we have deleted along the text any claim to a better motor development in Patient 2 (from the abstract and the discussion).

Point 6.

Growth and number of respiratory tract infections can also be used to determine the outcomes.

Response 6.

We agree with the reviewer´s suggestion and here we show a table with the anthropometrical data of the patients:

Age (months)

Weight (kg)

Weight percentile

Height (cm)

Height percentile

Patient 1

0

3.61

85

50

62

3.7

5.65

30

63

78

6.4

7.88

59

68

71

10.5

9.25

50

76

91

22

11.5

35

86

64

Patient 2

Age (months)

Weight (kg)

Weight percentile

Height

Height percentile

0.0

3.60

74

50.5

60

2.5

5.25

27

59

47

6.0

7.68

35

68

54

11

10.3

61

75

55

The following information has been added to the manuscript:

“3.3.3. Growth, respiratory and gastrointestinal status.

Both patients have presented normal growth, following their typical curves for height and

weight, within the normal range for age and sex.

Both patients have been asymptomatic, not requiring any respiratory support, and eating an age-appropriate diet by mouth with no assistance, and none of them has suffered any significant respiratory infection.”

DISCUSSION:

Point 7.

CRIM status is one of the important factors to determine the outcomes. Based on the genotypes, patient 1 was likely having CRIM negative and CRIM status of patient 2 was undetermined. Although I agreed that the resolution of patient’s HCM and biomarkers were better than patient 1’s, the fact that their genotypes were different should be mentioned as one of the limitations of this study.

Response 7.

We agree with the reviewer´s comment and state in the discussion “We also acknowledge that, besides age at treatment initiation, differences in treatment outcomes may be influenced by other factors such as the genotype, CRIM status, or residual enzymatic activity”.

Point 8.

Another limitation of this study, long term follow up is important since some patients who received prophylactic ITI and high dose ERT in the newborn or infancy periods may develop mild muscle weakness especially facial muscle in childhood period.

Response 8.

We agree with the reviewer´s comment and state: “Although in this study we present short-term follow-up, differences in the evolution of the two patients can be appreciated. Nevertheless, long-term narrow follow-up is warranted.”

Point 9.

Please clarify “A median time of 0.5 months has been reported from genetic and cross-reactive immunological material (CRIM) status” in line 225-227, what type of the study did the author refer to? Western blot CRIM study?

Response 9.

One important objective of our manuscript is to raise concern about the importance of timely treatment initiation in IOPD, and we show the different steps we followed in our center in the diagnosis and treatment initiation process (once IPD suspicion was very high, but before genetic confirmation, administrative procedures were initiated: an urgent meeting of the Pharmacy Committee that approves special treatments in rare diseases was set up, and rapid approval was obtained. Also, surgery for implantable venous catheter placement was scheduled).

To illustrate that it takes time to set up everything to start specific treatment once IOPD diagnosis has been made, in the discussion section we state: “A median time of 0.5 months has been reported from genetic and cross-reactive immunological material (CRIM) status diagnosis to commencement of ERT treatment, primarily due to administrative or regulatory issues (i.e., procurement of resources) in centres with expertise in Pompe disease -ref#6-”.

In this study (ref#6), “upon diagnostic confirmation of IPD, CN status was rapidly inferred by mutation analysis, using an established mutation database, which has allowed prediction of CN status in more than 90% cases. CRIM negative status was further confirmed using western blot analysis on skin fibroblast cell extracts, if none of the GAA protein processing forms (unprocessed precursor band at 110 kDa or processed forms bands at 95, 76 and 70 kDa) were detected”.

In this study, once CRIM status was determined, median time to commencement of ERT and ITI was 0.5 months (range: 0.1–1.6 months).

Point 10.

Line 254, the authors compared the anti-rhGAA antibodies after ITI of their 2 patients by using data from ref #20. Did the study group from ref#20 receive ITI? In theory, patients who received ITI should not develop high anti-rhGAA antibodies.

Response 10.

The reviewer is right and patients from ref#20 had not received ITI. We have completed the paragraph adding this information and also comparing to the evolution of IPD patients who received ITI treatment in ref#6:

“Our patients have tolerated ITI without any adverse events. It has been reported that most IPD patients develop anti-rhGAA antibodies at a median age of 1.8 months in the absence of ITI treatment (20). Another study in IPD patients who received prophylactic ITI treatment showed that 4/7 patients remained antibody free, and 3/7 patients developed antibodies at 23, 31 and 39 weeks after treatment initiation, 2 of them requiring another course of the ITI regimen (6). Our patients have not developed anti-drug antibodies so far.”

Point 11.

To compare the outcomes of these 2 patients with the literature, besides age of patients when the treatment starts, the other factors should be comparable e.g. CRIM status, ERT dosage (40MK weekly vs EOW vs 20MK), prophylactic ITI vs therapeutic ITI and whether the patients developed anti-rhGAA antibodies after ITI.

Response 11.

One of the messages of this paper is that we decided to treat with prophylactic ITI treatment irrespective of CRIM status, as suggested in ref#18. We decided this because some CRIM (+) patients -who would not receive ITI treatment according to the general suggestion of treating only CRIM (-) patients- have been reported to have developed high anti-rhGAA antibodies and have died consequently.

Apart from the age at treatment initiation (which is a very important prognosis factor), the two patients of this study are perfectly comparable as they have received exactly the same treatment regimen, and the same clinical and analytical follow-up. None of them has developed anti-rhGAA antibodies. Regarding CRIM status, it is undetermined according to their genotype in both patients.

Reviewer 2 Report

The Authors provided a description of clinical, biochemical and molecular data of 2 patients diagnosed with classic infantile-onset Pompe disease. The main limitation of this study is a weak clinical presentation of cases.

I propose to shorten the manuscript (especially discussion - it is too long) and categorization to a brief report.

Some Figures could be added to supplementary material or presented as one figure (if possible). I propose to transfer the Figure 1 as well as Table 1 to the supplementary data.

Please, find below some suggestions for improvement.

Page 3, verse 111-116: Some of the informations should be omitted. I propose to shorten the text: ,,At 2 months of age, a heart murmur and subtle hepatomegaly were observed''.

Page 3, verse 121: Please, provide the results of enzymatic analysis.

Page 3, verse 123: Besides of the word mutation, the Authors should use the word: variant/pathogenic variant, as recommended by the current guides on Human Genetics Societies.

Page 3, verse 129: Please, define the character of hyperbilirubinemia: conjugated or unconjugated.

Page 4, verse 132: Please, provide the results of enzymatic analysis.

Regarding Patient 2 - What about liver ALT, AST and CK ?

Regarding Patients 1 and 2 - What about cardiomegaly, feeding difficulties, failure to thrive ?

Regarding Patients 1 and 2 - Please, provide the results of neurolgical examination (not only in the context of hypotonia).

Page 5, verse 154: Similarly to above-presented Figures, please, provide the exact growth curves.

Final decision: Reconsider after major revision.

Author Response

Thank you very much for your review and comments. Our point-by-point response to each comment is detailed below.

The Authors provided a description of clinical, biochemical and molecular data of 2 patients diagnosed with classic infantile-onset Pompe disease. The main limitation of this study is a weak clinical presentation of cases.

Point 1.

I propose to shorten the manuscript (especially discussion - it is too long) and categorization to a brief report.

Some Figures could be added to supplementary material or presented as one figure (if possible). I propose to transfer the Figure 1 as well as Table 1 to the supplementary data.

Response 1.

The objective of the study was to analyse the different stages in the diagnosis and specific treatment initiation procedure in two IPD patients followed at the same Metabolic Unit and to compare clinical and biochemical outcomes depending on age at ERT initiation (<1 month of age vs. <3 months of age).

We believe that it is important and kindly ask to maintain the length of the manuscript to explain the different steps we followed in the diagnosis and treatment initiation procedure, to explain the different decisions that were made about the treatment of our patients (ITI treatment irrespective of CRIM status, high dose of ERT from the beginning), and to put in context the evolution of our patients comparing to the literature.

Please, find below some suggestions for improvement.

Point 2.

Page 3, verse 111-116: Some of the informations should be omitted. I propose to shorten the text: ,,At 2 months of age, a heart murmur and subtle hepatomegaly were observed''.

Response 2.

The paragraph has been shortened according to the reviewer´s suggestion.

Point 3.

Page 3, verse 121: Please, provide the results of enzymatic analysis.

Response 3.

The results of the enzymatic analysis have been included according to the reviewer´s suggestion.

Point 4.

Page 3, verse 123: Besides of the word mutation, the Authors should use the word: variant/pathogenic variant, as recommended by the current guides on Human Genetics Societies.

Response 4.

The text has been changed according to the reviewer´s suggestion.

Point 5.

Page 3, verse 129: Please, define the character of hyperbilirubinemia: conjugated or unconjugated.

Response 5.

The patient presented unconjugated hyperbilirubinemia, and the information has been added accordingly to the manuscript.

Point 6.

Page 4, verse 132: Please, provide the results of enzymatic analysis.

Response 6.

The results of the enzymatic analysis have been included according to the reviewer´s suggestion.

Point 7.

Regarding Patient 2 - What about liver ALT, AST and CK ?

Response 7.

The information has been added according to the reviewer´s suggestion.

Point 8.

Regarding Patients 1 and 2 - What about cardiomegaly, feeding difficulties, failure to thrive?

Response 8.

In the presentation of both clinical cases, we have added that the children were asymptomatic.

In the evolution, we have added the paragraph:

Both patients have presented normal growth, following their typical curves for height and weight, within the normal range for age and sex. Both patients have been asymptomatic, not requiring any respiratory support, and eating an age-appropriate diet by mouth with no assistance, and none of them has suffered any significant respiratory infection.

Point 9.

Regarding Patients 1 and 2 - Please, provide the results of neurolgical examination (not only in the context of hypotonia).

Response 9.

The information has been added according to the reviewer´s suggestion.

Point 10.

Page 5, verse 154: Similarly to above-presented Figures, please, provide the exact growth curves.

Response 10.

Both patients presented a normal growth following their typical curves for height and

weight, within the normal range for age and sex. This is what we would expect, as they are asymptomatic, with a normal psychomotor development without hypotonia and without any feeding difficulties.

Here we show a table with the anthropometrical data of the patients:

Age (months)

Weight (kg)

Weight percentile

Height (cm)

Height percentile

Patient 1

0

3.61

85

50

62

3.7

5.65

30

63

78

6.4

7.88

59

68

71

10.5

9.25

50

76

91

22

11.5

35

86

64

Patient 2

Age (months)

Weight (kg)

Weight percentile

Height

Height percentile

0.0

3.60

74

50.5

60

2.5

5.25

27

59

47

6.0

7.68

35

68

54

11

10.3

61

75

55

As the manuscript has already a lot of information and figures, we propose to add the following paragraph into the manuscript to give the information on growth:

3.3.3. Growth, respiratory and gastrointestinal status.

Both patients have presented normal growth, following their typical curves for height and

weight, within the normal range for age and sex.

Round 2

Reviewer 1 Report

no further comment

Reviewer 2 Report

The Authors reviewed their manuscript according to my suggestions.

Now, I recommend to publish the manuscript.

This manuscript is a resubmission of an earlier submission. The following is a list of the peer review reports and author responses from that submission.